# Predicting the behavior of AI agents using transfer operators

## Abstract

Predicting the behavior of AI-driven agents is particularly challenging without a preexisting model. In our paper, we address this by treating AI agents as stochastic nonlinear dynamical systems and adopting a probabilistic perspective to predict their statistical behavior using the Fokker-Planck equation. We formulate the approximation of the density transfer operator as an entropy minimization problem, which can be solved by leveraging the Markovian property and decomposing its spectrum. Our data-driven methodology simultaneously approximates the Markov operator to perform prediction of the evolution of the agents and also predicts the terminal probability density of AI agents, such as robotic systems and generative models. We demonstrate the effectiveness of our prediction model through extensive experiments on practical systems driven by AI algorithms.

## 1 Introduction

Autonomous agents operate in dynamic environments, making decisions based on continuous feedback that enables them to learn and adapt over time. Therefore, studying the behavior and alignment of these AI-driven agents is critical for several reasons. Analyzing their actions can help prevent behaviors that conflict with human values and ethical standards [Rossi & Mattei (2019), Doshi & Gmytrasiewicz (2005)]. Furthermore, understanding their behavior is essential for enhancing their efficiency and reliability, particularly in safety-critical applications such as autonomous robots [Pourmehr & Dadkhah (2012)]. These intelligent models are typically complex, high-dimensional, and only partially observable over short time intervals. This complexity raises the question of which properties can be efficiently quantified to truly understand their capabilities. The design and understanding of AI components embedded within these agents depend crucially on analyzing the interplay between AI-driven decision-making and the physical behavior of the closed-loop system. This capability is foundational for users to perceive, predict, and interact effectively with intelligent systems. It also provides the theoretical and technical basis for practical tasks such as decision-making and reinforcement learning [Levine et al. (2016), Ganzfried & Sandholm (2011)].

Given these challenges, it is important to develop methods capable of harnessing critical information to identify the behavior of AI components in closed-loop systems. In Déletang et al. (2021) and Roy et al. (2022), the authors try to model the AI agent as a system that was pre-trained using reinforcement learning and the environment is a partially-observable Markov decision process. The authors in Dipta et al. (2022) present a card game and leverage the game theory framework to analyze learning agents' characteristics with environmental changes. Moreover, Lee & Popović (2010) try capture a rich set of behavior variations by determining the appropriate reward function in the reinforcement learning framework. For a comprehensive review of literature on modeling and predicting AI agents behavior, we refer readers to Albrecht & Stone (2018). Among these emerging methodologies, there has been a notable increase in modeling these behaviors as nonlinear dynamical systems [Narendra & Parthasarathy (1992); Beer (1995); Suttle et al. (2024); Ijspeert et al. (2002)]. Originating from studies in partial differential equations (PDEs) and fluid mechanics, techniques such as Dynamic Mode Decomposition (DMD) and its generalizations have also demonstrated significant capability in revealing the underlying evolutionary laws of AI agents [ Schmid (2022), Brunton et al. (2021)].

Despite these advances made in the above mentioned works chaotic behavior resulting from nonlinearity and stochasticity encountered in practical problems pose significant challenges to deterministic

modeling and identification. However, despite the substantial uncertainties, and sensitivities to initial conditions affecting AI agents, their statistical behavior and properties can be surprisingly regular. Specifically, when the closed-loop dynamics of AI agents are time-invariant, the density evolution of the agents' states follows a Markovian property [Risken (1996), Gardiner (2009), Hoehn et al. (2005)].

These observations motivate analyzing this problem from a statistical perspective. In particular, modeling the evolution trajectories of the state via a stochastic process and then studying the transfer and propagation of the probability density functions, has gathered a lot of attention. This approach, drawing from statistical mechanics, assumes that the agents trajectories are independent and governed by the same Fokker-Planck equation, which characterizes the statistical behavior of agents based on their underlying dynamics, showing significant advantages in handling systems with complex, high-dimensional nonlinear chaotic dynamics and noise perturbations [Risken (1996); Lasota & Mackey (2013)]. Another motivation for analyzing the evolution of densities arises from generative AI. The characteristics of the sampling probability density are often intricate. Traditional statistical models frequently fall short due to the high dimensionality of the data and the inherent complexities of the sampling process. Generative models, such as those based on denoising diffusion processes [Ho et al. (2020)] or iterative reward-based sampling methods using transformers [Kingma (2013)], introduce complex stochastic dynamics. These dynamics are challenging to analyze at the level of individual samples. Instead, adopting a macroscopic perspective and focusing on the evolution of probability densities induced by these models enables the study of the aggregate behavior of complex models and a better understanding of their underlying mechanisms.

The application of probabilistic models to learn and predict the statistical behavior of complex AI agents has gained increasing attention in areas such as autonomous driving, motion planning, and human-robot interaction [Lefèvre et al. (2014); Trautman et al. (2015); Bai et al. (2015); Rasouli et al. (2017)]. However, algorithms based on this probabilistic perspective, particularly those studying the propagation of density processes, remain underexplored. Several open challenges persist in advancing this field, especially in developing unified frameworks that bridge the gap between domain-specific methodologies and general modeling approaches [Baarslag et al. (2016)].

In this work, we focus on virtual or physical agents with stochastic dynamics, which are controlled or operated by AI algorithms. We model the evolution of their state distributions as a Markov chain (or process). Unlike existing approaches that primarily predict agent evolution, our work adopts a macroscopic/statistical mechanics perspective. Our aim is to develop algorithms capable of predicting not only the propagation of future densities but also the stationary distribution of the Markov process, which corresponds to the controlled objective applied to the AI agents.

Specifically, we utilize the spectral decomposition theorem [Lasota & Mackey (2013)] for Markov operators to decompose the propagation of the Markov transfer operator into a transient decaying term and projections onto a set of cyclical bases representing the asymptotic behavior. As demonstrated in our analysis, this approach significantly simplifies the learning and representation of Markov processes, while also facilitating the prediction of the agents' future behavior from a macroscopic perspective.

## 1.1 RELATED WORK

The behavior prediction of AI-driven agents has been a growing focus in various domains, including robotics, generative modeling, and reinforcement learning. A number of approaches have been proposed, but they differ significantly in methodology, interpretability, and scope. Below, we situate our work within these efforts, identifying gaps that our approach aims to address.

### 1.1.1 STATISTICAL MODELING OF AI AGENTS

Modeling AI agents as stochastic systems has been explored in works like Goswami et al. (2018), who use constrained Ulam Dynamic Mode Decomposition to approximate Perron-Frobenius operators for deterministic and stochastic systems. Norton et al. (2018) proposes finite volume methods for numerical approximation, which form the foundation for deep learning-based operator approximation methods, such as DeepONets and Fourier Neural Operators [Li et al. (2021); De Ryck & Mishra (2022)]. However, these methods prioritize scalability and numerical efficiency, often at the cost of interpretability and robustness when applied to systems with high-dimensional chaotic dynamics.

Our approach addresses this gap by adopting a spectral decomposition framework, which not only approximates the Markov operator but also provides interpretable insights into the evolution and asymptotic behavior of densities.

### 1.1.2 BEHAVIOR PREDICTION AND REACHABILITY ANALYSIS

Several works have examined the behavior of AI agents through reachability analysis. Meng et al. (2022) directly learn state transfer functions, while Everett et al. (2021) and Zhang et al. (2023) analyze neural network-controlled systems to estimate their reachable states. These approaches focus on trajectory-specific predictions, which can become computationally expensive for systems with high-dimensional state spaces or stochastic dynamics.

Our method departs from these by adopting a macroscopic perspective: rather than tracking individual trajectories, we model the evolution of probability densities. This provides computational advantages and insights into the statistical behavior of systems without requiring exhaustive trajectory-level analysis.

### 1.1.3 GENERATIVE MODELING AND DIFFUSION PROCESSES

Generative models have also seen increasing adoption in behavior prediction, particularly through diffusion-based approaches [Song et al. (2020); Ho et al. (2020)]. These methods focus on sampling from complex distributions by iteratively refining noisy samples, but they do not explicitly model the density evolution over time. This makes it challenging to analyze the long-term statistical behavior of these models.

In contrast, our work incorporates diffusion modeling within a broader framework of Markov transfer operators, enabling both the prediction of density evolution and the estimation of stationary distributions.

### 1.1.4 LIMITATIONS OF EXISTING APPROACHES

Despite their contributions, many existing methods are either limited in scope or fail to generalize across diverse application domains. For instance:

- Reachability methods often require exhaustive computation of individual trajectories, which is infeasible for high-dimensional stochastic systems.
- Generative models excel at producing samples but do not explicitly address density evolution, limiting their applicability for long-term behavior prediction.
- Operator approximation methods prioritize computational efficiency but lack interpretability or alignment with asymptotic properties.

### 1.2 OUR CONTRIBUTIONS

Our main contributions are detailed as follows:

- AI-driven agents behave in unpredictable ways due to machine learning black boxes. We look at this through the lens of propagation of probability densities and the stochastic transfer operator. AI agents are trained with data that has inherent biases. This, coupled with the structure of machine learning models, can potentially alter the alignment of the model. To verify the alignment of the model, we predict the asymptotic behavior of the model by analyzing the terminal stationary density of the AI agents.
- We propose PISA, a novel and scalable algorithm that can simultaneously predict the evolution of the densities of AI agents and estimate their terminal density. Our algorithm is motivated by the spectral decomposition theorem [Lasota & Mackey (2013)] and provides a theoretical backing for its performance. PISA simultaneously approximates the action of the Markov transfer operator from the trajectory data of agents and predicts their asymptotic behavior.
- In our proposed algorithm PISA, the model complexity is indexed by the number of basis functions. The number of basis functions is a tunable parameter that can be altered according

to the user's needs. We provide a theoretical guarantee of the existence of the optimal solution to our operator estimation problem.

- We numerically verify the effectiveness of PISA in a variety of practical cases and compare it with existing literature. We first predict the behavior of unicycle robots driven by a controller based on diffusion models. Then we analyze the behavior of generative models from the lens of density evolution. Lastly, we apply PISA in the case of predicting the movement of pedestrians. We observe that PISA performs significantly better than the existing literature.

Briefly speaking, by adopting a spectral decomposition approach, our proposed method bridges the gaps in these prior works. Specifically:

- It balances computational efficiency with interpretability, providing insights into both short-term evolution and long-term stationary behavior.

- It generalizes to diverse systems, including high-dimensional chaotic systems, by focusing on density evolution rather than individual trajectories.

- It integrates with advanced generative models and reinforcement learning frameworks, providing a unified approach to behavior prediction.

This contextualization positions our work as a step toward a more comprehensive framework for understanding the statistical behavior of AI-driven agents.

## 1.3 STATISTICAL BEHAVIOR PERSPECTIVE AND THE FOKKER-PLANCK EQUATION

Consider a practical AI agent with physical dynamics defined by

$$\dot{x} = h(x, u) + g(x)\xi, \tag{1}$$

where $u$ is an external input to the system and $\xi$ represents the white noise signal. With a parameterized machine learning model as feedback $u = \texttt{ML}_\theta(x)$, the system's dynamics including the feedback input is given by

$$\dot{x} = h(x, \texttt{ML}_\theta(x)) + g(x)\xi = f(x) + g(x)\xi, \tag{2}$$

where $x(t) \in X \subseteq \mathbb{R}^M$ and $f(\cdot) : \mathbb{R}^M \mapsto \mathbb{R}^M$ and $g(\cdot) : \mathbb{R}^M \mapsto \mathbb{R}^p$ are nonlinear continuous functions.

The nonlinear system (2) is highly dependent on initial conditions and the noise $\xi$. Instead of analyzing individual trajectories of (2), we take the perspective of analyzing several independent trajectories simultaneously. Despite the challenges posed by stochasticity, and complexity in the system dynamics, the evolution of the statistical distribution over the states of all agents remains well-structured [Lasota & Mackey (2013)]. Particularly, at each time instance, samples from all the independent trajectories can be viewed as a probability density of the states. Therefore, the evolution of states from various initial conditions can be viewed as the evolution of a probability density. The evolution of the probability density function of states at time $t$, denoted by $\rho(x, t)$, forms a Markov process that obeys the Fokker-Planck equation, as described below:

**Lemma 1 (Risken (1996))** *For agents governed by (2), we have that the evolution of the density of states $\rho(x, t)$ is a Markov process. The evolution is given by the Fokker-Planck equation*

$$\frac{\partial \rho(x, t)}{\partial t} = -\sum_{i=1}^{n} \frac{\partial \left( f_i(x, u_e) \rho(x, t) \right)}{\partial x_i} + \frac{1}{2} \sum_{i=1}^{n} \sum_{j=1}^{n} \frac{\partial^2 \left( g(x) g^T(x) \rho(x, t) \right)_{ij}}{\partial x_i \partial x_j} = A_{FP} \circ \rho(x, t), \tag{3}$$

*where $A_{FP}$ is the differential operator the characterizes the evolution of $\rho(x, t)$ with time, also denoted as the Markov transfer operator. For the series of densities $\{\rho_k(x)\} = \{\rho(x, k\tau)\}$ with some $\tau > 0$, the density transfer operator $P$ is a Markov operator such that*

$$\rho_{k+1}(x) = P \circ \rho_k(x), \tag{4}$$

*and if the Markov process is constrictive [(Lasota & Mackey, 2013, Definition 5.3.1)], then there is a correspondent stationary density $\rho^*(x)$ such that*

$$\rho^*(x) = P \circ \rho^*(x). \tag{5}$$

**Remark 1** *It is important to note that the Markov transfer operator $P$ completely defines the evolution of the density of the system. Hence, our goal is to analyze the behavior of the AI-driven agents given by (2), through the estimation of the action of $P$. Our goal is to also estimate the asymptotic behavior of AI-driven agents as several systems (2) exhibit stationary states asymptotically. For example, robotics systems are designed to stabilize certain points in the domain. Another example is a diffusion model which is trained to sample from unknown target distributions. For systems that exhibit stationary states, there exists an invariant density $\rho^*$ [Lasota & Mackey (2013)] for the Markov operator such that (5) holds. Here agents following (2) reach $\rho^*$ asymptotically. We seek to estimate the terminal density $\rho^*$ as it provides a convenient method to assess the alignment of the AI-driven agents.*

**Remark 2** *We have noticed that for some general AI agents, the density evolution process of the state may not be a Markovian process. This happens, for example, when $h(x, u)$, $g(x)$ or $u(x)$ in (1) and (2) are time-variant and non-stationary. In these cases, a similar form of time-variant Fokker-Planck equation holds, though it does not implies a Markov process [Risken (1996)]. Nonetheless, we claim that the Markovian setting is actually widely used and considered for model simplicity and convenience of analysis. To give an example, the state evolution of Markov decision processes (MDPs) with any fixed (stationary) control policy applied, forms a Markov process [Bertsekas (2012)].*

### 1.4 FROM SAMPLES TO DENSITIES

Our data consists of the state trajectory of $N$ identical agents governed by the dynamics (2). The trajectory of these agents are collected from $t = 0$ to $t = T$ with a fixed sampling period $\tau = \frac{T}{K}$. The sampled dataset is given by $\{\mathcal{X}_n\}_{n=1}^N$ of the state $x$, where $\mathcal{X}_n = [\chi_0^n, \chi_1^n, \chi_2^n, \cdots, \chi_K^n] \in \mathbb{R}^{M \times (K+1)}$. Note that the collected data set can also come from a single agent starting from $N$ different initial states.

Estimating probability densities from samples is an active problem in machine learning and statistics. In this work, we employ Kernel Density Estimation to numerically construct the probability density $\rho_k(x)$ using the data $\{\mathcal{X}_n\}$. We can view $\{\mathcal{X}_n\}$ by iterating with respect to time as $\{\mathcal{Y}_k\}_{k=0}^K$, where $\mathcal{Y}_k = [\chi_k^1, \chi_k^2, \cdots, \chi_k^N]$ denotes the state vectors of $N$ particles at time $t = k\tau$. Using kernel density estimation [Hastie et al. (2009)], we then get an empirical probability distribution estimation $\rho_k(x)$. In this paper, we choose to use the Gaussian kernel for the estimation of $\rho_k$ which is given by

$$\rho_k(x) = \frac{1}{N\sqrt{\det(2\pi\sigma_k^2 I_M)}} \sum_{n=1}^N e^{-\frac{\|x - \chi_k^n\|^2}{2\sigma_k^2}}. \tag{6}$$

It is important to note that any choice of density estimation algorithm can be used with the data $\{\mathcal{X}_n\}$ to obtain $\{\rho_k(x)\}_{k=0}^K$. KDE provides a convenient choice for measuring the probability $\rho(x)$ at fixed reference points. The choice of reference points and the parameter $\sigma_k$ can be chosen by the user to better approximate $\rho_k$. In this paper, we choose to uniformly sample the reference points in the domain $X$ to estimate every $\rho_k(x)$. We fix a constant $\sigma_K = \sigma$ for simplicity. More specific KDE-related tools can be used based on the system in consideration and the domain, as enlisted in [Chen (2017)].

We illustrate in Figure 1 how the state trajectory $x(t)$ is coupled with the probability density $\rho(x, t)$ for the Van der Pol oscillator,

$$\dot{x}_1 = x_2, \quad \dot{x}_2 = \mu(1 - x_1^2)x_2 - x_1.$$

**Remark 3** *We acknowledge that the choice of kernels significantly affects both the accuracy and the computational complexity of the algorithm. However, since our main focus lies in predicting the trajectories of densities, we opted for a practical kernel choice tailored to the application at hand. More sophisticated methods to approximate densities from data choosing might yield better results and our algorithm can easily be used with these techniques.*

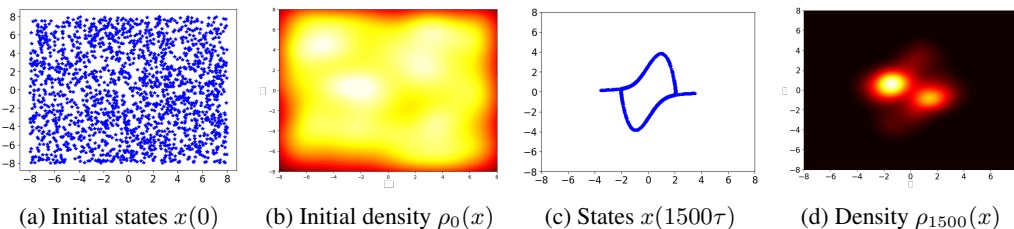

(a) Initial states $x(0)$     (b) Initial density $\rho_0(x)$     (c) States $x(1500\tau)$     (d) Density $\rho_{1500}(x)$

Figure 1: Illustration of the relationship between the states and probability density of the Van der Pol oscillator in a bounded domain. The x-axis in the figures corresponds to $x_1$ and the y-axis in the figures corresponds to $x_2$. (a) Various particles with different initial states driven by Van der Pol dynamics. (b) Density of initial state of agents. (c) States of the agent after time $t = 1500\tau$. (d) Density of the states at time $t = 1500\tau$. Brighter colors in (b) and (d) represent higher probability. The states are sampled with $t = 0.01$.

## 2    PREDICTION INFORMED BY SPECTRAL-DECOMPOSITION ALGORITHM (PISA) FOR LEARNING MARKOV TRANSFER OPERATORS

We present our algorithm to estimate the Markov transfer operator in this section. Further, we predict the asymptotic behavior of the system by estimating the terminal density of the dynamical system. We approximate the action of the Markov transfer operator using the following model,

$$P \circ \rho_k(x) = \frac{1}{l} \sum_{i=1}^{l} \rho_{k-l+i}(x) - \sum_{i=1}^{l} \left( \frac{1}{l} - \mathtt{A}_\theta^i(\rho_k) \right) \mathtt{G}_\gamma^i(x). \tag{7}$$

Here, we are decomposing the action of the Markov transfer operator on the density $\rho_k(x)$ into $2l$ components as given by $l$ functionals $\mathtt{A}_\theta^i(\rho_k)$ and $l$ functions $\mathtt{G}_\gamma^i(x)$. The functionals $\mathtt{A}_\theta^i(\rho_k)$ and functions $\mathtt{G}_\gamma^i(x)$ are parameterized by $\theta$ and $\gamma$ respectively. Moreover, we impose the following constraints:

$$P \circ \mathtt{G}_\gamma^i(x) = \mathtt{G}_\gamma^{i+1}(x), \quad \forall i = 1, \cdots, l-1;$$
$$P \circ \mathtt{G}_\gamma^l(x) = \mathtt{G}_\gamma^1(x); \tag{8}$$
$$\langle \mathtt{G}_\gamma^i(x), \mathtt{G}_\gamma^j(x) \rangle = 0, \forall i \neq j.$$

This method of decomposing the Markov transfer operator is guided by the spectral decomposition theorem which we elaborate on in Section 4.

Given the decomposition of the Markov transfer operator, we propose the following loss function, guided by the spectral decomposition theorem, to learn the parameter $\theta$ and $\gamma$.

$$L(\theta, \gamma) = \sum_{k=l-1}^{K-1} D \left( \frac{1}{l} \sum_{i=1}^{l} \rho_{k-l+i}(x) - \sum_{i=1}^{l} \left( \frac{1}{l} - \mathtt{A}_\theta^i(\rho_k) \right) \mathtt{G}_\gamma^i(x) \middle\| \rho_{k+1}(x) \right)$$
$$+ \lambda \sum_{i \neq j}^{l} \langle \mathtt{G}_\gamma^i(x), \mathtt{G}_\gamma^j(x) \rangle + \mu \sum_{r=1}^{l} D \left( \sum_{i=1}^{l} \mathtt{A}_\theta^i(\mathtt{G}_\gamma^r) \mathtt{G}_\gamma^i(x) \middle\| \mathtt{G}_\gamma^{r+1}(x) \right) \tag{9}$$

We then construct PISA as the following alternating optimization algorithm to compute $\theta$ and $\gamma$, in which we choose $\mathtt{A}_\theta^i(\rho)$ and $\mathtt{G}_\gamma^i(x)$ to be outputs of two distinct neural networks parameterized by $\theta$ and $\gamma$, respectively. An important aspect of PISA is that it can also predict the terminal density of the Markov transfer operator. The estimate of terminal density of $P$ can be expressed as

$$\rho^*(x) = \frac{1}{l} \sum_{i=1}^{l} \mathtt{G}_\gamma^i(x). \tag{10}$$

Note that optimization problems (11) and (12) are constrained by the structure of $\mathtt{G}_\gamma^i$ and $\mathtt{A}_\theta^i$. These constraints are easily satisfied by non-negative output layers of typical neural network architectures. For instance, normalized sigmoid layer for $\mathtt{G}_\gamma^i$ satisfies the constraints in (11) and (12).

---

**Algorithm 1: Prediction Informed by Spectral-decomposition Algorithm (PISA)**

---

**Data:** $l > 0$, $\lambda > 0$, $\mu > 0$; $\rho_k(x)$, for $k = 0, 1, \cdots, K$; initial values of $\gamma$ and $\theta$; two small
      positive thresholds $\epsilon_1$ and $\epsilon_2$;

**Result:** $\gamma$ and $\theta$;

1   $N_{\text{epochs}} \leftarrow 1000$

2 **while** $N_{epochs} \neq 0$ **do**

3     Solve the following optimization problem to get $\gamma^*$

$$\min_{\gamma} \quad L(\theta, \gamma)$$

$$\text{s.t. } \mathsf{G}^i_\gamma(x) \geq 0 \text{ and } \int \mathsf{G}^i_\gamma(x)dx = 1, \text{ for } i = 1, \cdots, l; \tag{11}$$

4     **if** $\|\gamma^* - \gamma\| \geq \epsilon_1$ **then**

5        $\gamma \leftarrow \gamma^*$

6     **end**

7     Solve the following optimization problem to get $\theta^*$

$$\min_{\theta} \quad L(\theta, \gamma)$$

$$\text{s.t. } \mathsf{A}^i_\theta(\rho_k) \geq 0, \text{ for } i = 1, \cdots, l \text{ and } \sum_{i=1}^{l} \mathsf{A}^i_\theta(\rho_k) = 1; \tag{12}$$

8     **if** $\|\theta^* - \theta\| \geq \epsilon_2$ **then**

9        $\theta \leftarrow \theta^*$

10    **end**

11    $N_{\text{epochs}} = N_{\text{epochs}} - 1$

12 **end**

---

## 3   Numerical Experiments

We present the effectiveness of PISA on different numerical testbeds. We performed the numerical experiments on a machine with Intel i9-9900K CPU with 128GB RAM and the Nvidia Quadro RTX 4000 GPU. In our numerical experiments, we compare the performance of PISA with that of Meng et al. (2022) and DDPD proposed in Zhao & Jiang (2023a). Particularly, Meng et al. (2022) approximates the Perron-Frobenius operator as

$$\rho_{k+1} = e^{t \cdot \text{NN}_\delta(x,t)} \rho_k.$$

Here, note that $\text{NN}_\delta$ approximates the differential operator $A_{FP}$ given in (3). Then $e^{t \cdot \text{NN}_\delta}$ is an approximately linear solution to (4). Moreover, in Zhao & Jiang (2023a), the authors provided the dynamic probability density decomposition (DPDD) method, which is based on Extended Dynamic Mode Decomposition and makes a linear finite-dimensional approximation of the Markov transfer operator to forecast the density evolution.

### 3.1   Lunar Lander (Continuous)

We apply PISA to predict the behavior of a reinforcement learning algorithm. Lunar lander (Continuous) is a rocket trajectory optimization problem on the Gymnasium platform Towers et al. (2024), with an eight-dimensional state and three-dimensional control input. We first train a feedback control policy using the Actor-Critic algorithm to make the rocket land on the landing pad which is always given by the coordinates (0,0) in the simulation environment. The feedback policy results in stochastic nonlinear dynamics as described in (2). We collect 3000 trajectories of length 500 time steps. We evaluate the density of the states via kernel density estimation (KDE) method to get a trajectory of densities as $\{\rho_t(x)\}_{t=0}^{500}$. We use the first 100 steps of the density trajectory to learn the Markov transfer operator using PISA. We also estimate the Markov transfer operator using DPDD (Zhao & Jiang (2023a)), and direct NN (Meng et al. (2022)), respectively to compare the different models. We evaluate the performance of the learned models to predict the density for the following 400 steps to

get $\{\hat{\rho}_t(x)\}_{t=101}^{500}$ for each of the three algorithms. We use the KL divergence between the predicted density $\hat{\rho}_k$ and the true density $\rho_k$ to characterize the performance of each algorithm. In Figure 2(a), we compare the performance of the three algorithms. We see that Meng et al. (2022) performs better initially due to its linearity assumption on the transfer operator but PISA performs better in the long term. However, DPDD performs significantly worse as it projects the operator on a finite basis. In Figure 2(b), we depict the predicted stationary density $\rho^*$ of the lunar lander. We see that $\rho^*$ is centered around (0,0) with a high probability which verifies that the controller makes the rocket land within the landing pad most times.

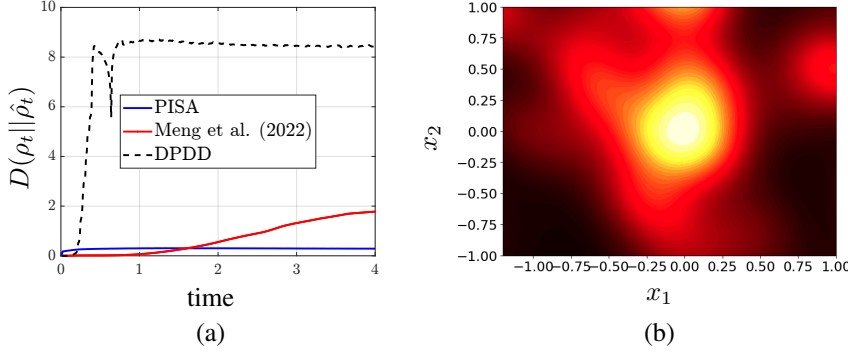

Figure 2: Experiments on the lunar lander in the Gymnasium environment. (a) Comparison of performance between PISA, Meng et al. (2022) and DPDD. PISA performs better than the other algorithms in the long term although Meng et al. (2022) performs better initially due to its linearity assumption. (b) Prediction of the stationary density $\rho^*$ of the lunar lander centered at (0,0). This confirms that the controller ensures that the rocket lands within the landing pad.

### 3.2 PREDICTING BEHAVIOR OF SCORE-BASED GENERATIVE MODEL

We consider the problem of analysis of the behavior of diffusion models. Diffusion models generate data using a bidirectional scheme. Given data samples from an unknown density, in the forward process, noise is sequentially added until the data samples resemble white noise samples from a standard Gaussian. Then, to generate new samples, the reverse process of diffusion models iteratively removes noise from white noise samples to generate realistic samples from the target density. The reverse process is particularly complex as the amount of noise to be removed at every step is estimated using neural networks. Our task is to analyze the behavior of diffusion models in the reverse process from the lens of evolving probability densities. We particularly consider the case of diffusion models based on estimating the score Song et al. (2020). We seek to study the behavior of these score-based generative models by their action on samples in the reverse process.

In Song et al. (2020), the forward and reverse processes use the following Stochastic Differential Equation (SDE),

$$\dot{x} = Ax + Bu, \tag{13}$$

where $x, u \in \mathbb{R}^{10}$. In the forward process where noise is sequentially added, $x(0)$ are samples from an unknown target distribution, and $u(t)$ is sampled from a Gaussian $u(t) \sim \mathcal{N}(\mathbf{0}, I)$. In the reverse process, $x(T) \sim \mathcal{N}(\mathbf{0}, I)$ is the initial point, and $u(t)$ is the output of a neural network $\mathsf{S}_\psi(x, t)$ that estimates the amount of noise needed to be removed at each step $t$ to obtain a realistic sample. In this experiment, we predict the behavior of the score-based diffusion model in the reverse process. For the reverse process, the task is to sample from Gaussians centered at $-6\mathbf{1}$ and $6\mathbf{1}$ and $k\boldsymbol{I}$ where $k \sim \mathcal{N}(-3, I)$. The noise distribution is the standard Gaussian. In Figure 3(a) we show the first two dimensions of the samples used in the reverse process of the diffusion model. The blue points denote the initial points sampled from a standard Gaussian. The red points denote the final samples from the target distribution. To train the diffusion model, we use $N = 12000$ samples from the target distribution. The data samples are diffused in the forward process for a time period of 8 seconds. In the reverse process, to sample from the desired distributions, we learn the score as proposed in Song et al. (2020). Once the score is sufficiently learned using a neural network, we record $N$ trajectories

in the reverse process for a time period of five seconds, which constitute the training dataset. Then we predict the behavior for the next three seconds which is the testing dataset.

We use the KL divergence between the predicted $\hat{\rho}_t$ and the true $\rho_t$ from the testing dataset as a metric to numerically analyze the performance of PISA. It is evident from Figure 3(b) that PISA accurately predicts the behavior of the diffusion model with varying choices of the number of basis functions. We use the logarithm of the KL divergence to emphasize that PISA performs at least one order of magnitude better than the other methods Meng et al. (2022) and DDPD Zhao & Jiang (2023a). We see that Meng et al. (2022) performs better initially due to its linear solution to the PDE but its performance deteriorates rapidly. PISA retains relatively much better performance over a longer time horizon. In Figure 3(c), we take a closer look at the performance of PISA for different choices of basis functions $l$. We see that a choice of $l \geq 10$ is required to achieve good performance. We use feedforward neural networks with 3 hidden layers for $\mathbb{A}_\theta^i$, $\mathbb{G}_\gamma^i$ and $\mathbb{NN}_\delta^i$.

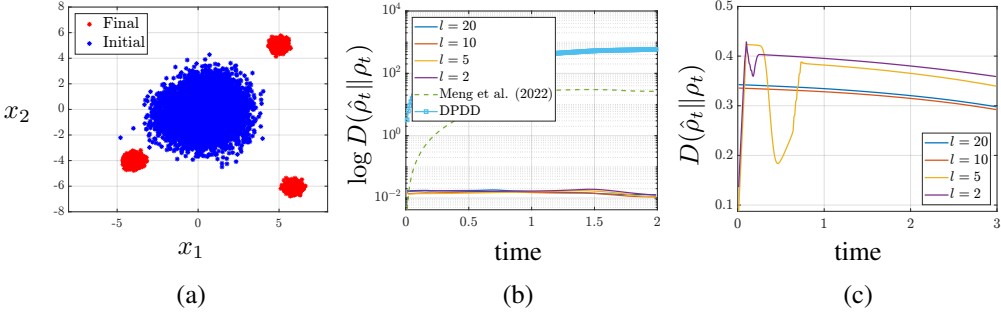

(a)               (b)               (c)

Figure 3: Experiments on the ten dimensional score-based generative model Song et al. (2020). (a) Data samples of diffusion model. Blue points of the standard Gaussian are the initial points in the reverse process. Red points denote samples from the unknown final distribution. (b) Comparison of performance of PISA with different choices of basis functions $l$, Meng et al. (2022) and DPDD on testing dataset. PISA performs an order of magnitude better for any choice of basis function. (c) Comparison of PISA with different choices of basis functions $l$. A choice of $l \geq 10$ is required to achieve the best performance.

### 3.3 UCY PEDESTRIAN DATASET

Here, we show the effectiveness of PISA on physical data where the transfer operator is not constrictive and the dynamics are not Markov. We apply PISA on the UCY pedestrian dataset Lerner et al. (2007) to predict the movement of pedestrians by estimating the evolution of the density of pedestrians. The dataset consists of videos of pedestrians walking in several regions as depicted in Figure 4(a). We use the Zara01 subsection of the dataset in our experiments. We obtained pre-processed data from the code repository of Salzmann et al. (2020), where the video was processed to obtain the $x$ and $y$ coordinates of the position of the pedestrians. As pedestrians are walking everywhere in state space, the constrictive property no longer holds. Further, pedestrians enter and exit the scene which results in highly stochastic behavior in the density. Given these complications, we show that PISA still performs better than other methods that learn transfer operators. We assume that every pedestrian is identical and their movement is governed by the dynamics given in (2). Both DDPD and Meng et al. (2022) require Markovian dynamics of the density, however we show that PISA can perform well even when this assumption fails.

Given the positions of pedestrians as depicted in Figure 4(a), we approximate the probability density of the pedestrians as depicted in Figure 4(b). In Figure 4(c), we once again compare PISA with the exponential model Meng et al. (2022) on the test data for the first 400 time samples. We choose $l = 5$ and feedforward neural networks with 3 hidden layers for $\mathbb{A}_\theta^i$, $\mathbb{G}_\gamma^i$ and $\mathbb{NN}_\delta^i$. Here, we see that the initial time period in which the exponential model works better than PISA is significantly shorter due to the model inaccuracy. However, PISA continues to perform well over a longer time horizon. It is also important to note that both models have significantly higher estimation errors in the testing performance for this experiment compared to performance in experiments on the Gymnaisum Lunar Lander model and the score-based generative model. This is due to the stochasticity of the data and the assumption we make about the nature of the pedestrians. Better density approximation

algorithms that are suited for stochastic data and for incorporating jumps in the probability density can be employed to obtain an improvement in the performance.

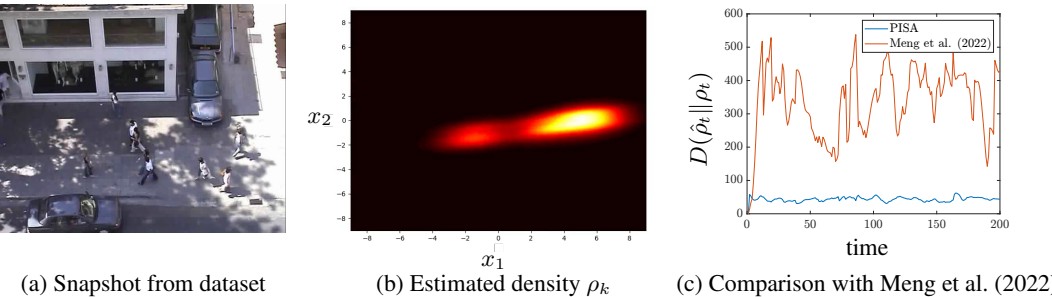

(a) Snapshot from dataset      (b) Estimated density $\rho_k$      (c) Comparison with Meng et al. (2022)

Figure 4: Experiments on the UCY pedestrian dataset. (a) A snapshot from the dataset. (b) Corresponding estimated probability density. (c) Comparison between PISA and Meng et al. (2022) in the estimation of future probability densities.

## 4 THEORETICAL FOUNDATIONS OF SPECTRAL DECOMPOSITION

If the Markov transfer operator $P$ is constrictive, it pushes forward the probability distribution $\rho_k(x)$ to a stationary distribution $\rho^*(x)$ corresponding to the attractors of dynamical systems. This evolution of the probability distribution is in fact a *Markov Process* (see Appendix A). For Markov operators with the constrictive property, we have the following spectral decomposition theorem.

**Lemma 2 (Lasota & Mackey (2013) )** *Let $P$ be a constrictive Markov operator. Then there exists an integer $l$, two sequences of non-negative functions $g_i(x) \in \mathcal{L}_1$ and $h_i(x) \in \mathcal{L}_\infty$, $i = 1, 2, \cdots, l$, and an operator $Q : \mathcal{L}_1 \mapsto \mathcal{L}_1$ such that for all $\rho(x) \in \mathcal{L}_1$, $P \circ \rho(x)$ can be written in the form*

$$P \circ \rho(x) = \sum_{i=1}^{l} a_i(\rho) g_i(x) + Q \circ \rho(x), \tag{14}$$

*where*

$$a_i(\rho) = \int \rho(x) h_i(x) dx.$$

*The functions $g_i(x)$ and the operator $Q$ have the following properties:*

*1) Each $g_i(x)$ is normalized to one and*

$$g_i(x) g_j(x) = 0, \quad \text{for all } i \neq j, \tag{15}$$

*i.e., the density functions $g_i(x)$ have disjoint supports;*

*2) For each integer $i$ there exists a unique integer $\alpha(i)$ such that*

$$P \circ g_i(x) = g_{\alpha(i)}(x). \tag{16}$$

*where $\alpha(i) \neq \alpha(j)$ for $i \neq j$. Thus, $P$ just permutes the functions $g_i(x)$;*

*3) Moreover,*

$$\|P^n Q \circ \rho(x)\| \to 0 \tag{17}$$

*as $n \to \infty$ for every $\rho(x) \in \mathcal{L}_1$.*

Lemma 2 states that the action of the PF operator can be decomposed into $l$ components through the functionals $a_i(\rho)$ and the functions $g_i(x)$. Here $l$ is a finite integer that serves as a measure of the model complexity of PISA. Further, the operator $Q$ captures the effect of the terminal density on $\rho(x)$. As $t \to \infty$, the action of $Q$ on $\rho(x)$ decays to 0. This drives our motivation to use $Q \circ \rho(x)$ as

$$Q \circ \rho_k(x) = \frac{1}{l} \sum_{i=1}^{l} \rho_{k-l+i}(x) - \frac{1}{l} \sum_{i=1}^{l} g_i(x). \tag{18}$$

Further, as $t \to \infty$, we can see that

$$\rho^*(x) = \frac{1}{l} \sum_i^l g_i(x). \tag{19}$$

This implies that the density functions $g_i$ serve as a basis for the stationary terminal density $\rho^*$. It is easy to verify for (19) that $P \circ \rho^*(x) = \rho^*(x)$ through the permutation property. Given the Lemma 2, (18), and (19), we provide a sufficient condition on the output of our algorithm PISA.

**Theorem 1** *For systems* (2) *that have a stationary terminal density, there exists a finite $l$, an operator $Q$, $l$ non-negative functionals $\mathbb{A}_\theta^i(\rho)$ and $l$ densities $\mathbb{G}_\gamma(x)$ such that the loss $L(\theta, \gamma) = 0$.*

**Proof 1** *We provide a brief overview of the proof. Lemma 2 guarantees the existence of $l$ functions $a_i(\rho)$ and $g_i(x)$ that exactly decompose the action of the PF operator. These are approximated using neural networks $\mathbb{A}_\theta^i(\rho)$ and $\mathbb{G}_\gamma^i(x)$, respectively. The cost function $L$ is designed to satisfy the properties of $a_i$ and $g_i$. The first term in the cost function $L$ addresses the propagation of the PF operator. The second term addresses the orthogonality property of every $g_i$ and the last term captures the permutative property $g_i$.* ∎

## 5 CONCLUSION AND BROADER DISCUSSION

While this paper introduces the PISA framework for predicting the behavior of AI-driven agents and demonstrates its advantages over existing methods, there remain several broader considerations regarding its applicability, interpretability, and limitations in the context of understanding AI systems.

### 5.1 COMPARISON WITH OTHER METHODS

PISA leverages spectral decomposition to approximate the Markov transfer operator, providing a theoretically grounded approach for both short-term prediction and asymptotic behavior estimation. Compared to alternative methods, such as direct neural network-based approximations (e.g., Meng et al. (2022)) or dynamic mode decomposition approaches (e.g., Zhao & Jiang (2023b)), PISA shows superior long-term prediction accuracy. However, these advantages are subject to the following trade-offs:

- **Interpretability:** The spectral decomposition framework provides a clearer interpretative advantage by explicitly decomposing density evolution into orthogonal components. In contrast, purely neural network-based methods, while often achieving strong performance, lack this interpretability due to their black-box nature.
- **Prediction Accuracy:** While PISA outperforms existing methods in long-term predictions, its accuracy is sensitive to the number of basis functions ($l$) used in the decomposition. This dependency introduces a trade-off between computational complexity and predictive fidelity, which needs careful tuning based on the specific task.
- **Flexibility in Non-Markovian Systems:** PISA assumes a Markovian setting for density evolution. Though it can perform well in approximately Markovian or partially observable systems (as shown in the UCY pedestrian dataset), its performance may degrade when the dynamics deviate significantly from this assumption.

### 5.2 KERNEL DENSITY ESTIMATION AND BROADER LIMITATIONS

The conclusion section highlights KDE as a potential limitation due to its dependency on kernel choice and bandwidth parameters. While KDE is computationally efficient and widely applicable, it introduces the following challenges:

- **Scalability in High Dimensions:** KDE struggles in high-dimensional settings due to the curse of dimensionality, which can limit its applicability to complex systems with large state spaces.
- **Sensitivity to Parameters:** The choice of kernel bandwidth directly impacts the quality of density estimates, necessitating careful tuning that may not generalize across tasks.

Alternative density estimation methods, such as normalizing flows or diffusion-based models, could be integrated into the PISA framework to mitigate these issues. However, such methods may increase computational overhead and reduce interpretability.

## 5.3 UNDERSTANDING AI SYSTEMS IN BROADER CONTEXT

The stated goal of this work is to provide tools to understand AI systems. While PISA contributes to this understanding by predicting density evolution and terminal behavior, several broader challenges remain:

- **Alignment with System-Level Interpretability:** The focus on density evolution offers macroscopic insights but may not fully address questions about individual trajectory behaviors or their implications for safety and alignment in AI systems.
- **Trade-offs Between Generality and Specificity:** PISA's general approach enables its application across diverse domains, but domain-specific adaptations (e.g., customized kernels, task-tailored loss terms) might be needed to achieve maximum predictive accuracy and insight.
- **Broader Metrics for Evaluation:** While KL divergence is used as the primary metric for performance evaluation, incorporating additional metrics, such as sensitivity to outliers, robustness to noise, or interpretability indices, could offer a more comprehensive assessment of the framework's effectiveness.

## 5.4 FUTURE DIRECTIONS

To fully situate PISA in the broader landscape of methods for understanding AI systems, future research could:

- Investigate hybrid approaches that combine the interpretability of spectral methods with the flexibility of neural network-based approximations.
- Explore non-Markovian extensions of PISA to capture more complex dynamics and interactions.
- Develop methods to quantify and compare interpretability and alignment performance across different prediction frameworks.
- Incorporate task-specific constraints, such as physical feasibility or safety guarantees, directly into the framework.

**In conclusion**, while PISA represents a significant step forward in modeling the statistical behavior of AI agents, its success in udnerstanding such systems requires continued exploration of these broader considerations. By addressing the interpretability, flexibility, and scalability of methods, future work can further contribute to the development of trustworthy AI systems.

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

## A   OPERATOR THEORY AND STATISTICAL MECHANICS

The study of density evolution from a macroscopic perspective is rooted in the principles of statistical mechanics, which bridge the microscopic behavior of individual components and the macroscopic properties of complex systems. Foundational works, such as those by Lasota & Mackey (2013) and Risken (1996), have established frameworks for examining how distributions of states evolve over time in systems governed by stochastic dynamics. This perspective is particularly valuable in systems with high-dimensional, nonlinear dynamics, such as AI-driven agents, where direct analysis of individual trajectories is impractical.

The macroscopic approach shifts focus from tracking each particle or agent to analyzing the aggregate behavior of a population. By modeling the evolution of probability densities, this framework enables predictions about the statistical behavior of the system as a whole, revealing regularities that emerge despite underlying stochasticity and chaos. A crucial simplification in this framework is the assumption of independence among trajectories.

**Assumption 1 (Ash (2012))** *(Independent Particles Approximation) In our basic problem setting, we assume that there are $N$ trajectories indexed by $n$ and each trajectory $\mathcal{X}_n = [\chi_0^n, \chi_1^n, \chi_2^n, \cdots, \chi_K^n] \in \mathbb{R}^{M \times (K+1)}$ is generated independently and governed by the identical systems dynamics as shown in (2).*

This assumption, inspired by the ideal gas model in thermodynamics, allows the collective behavior of the system to be captured using a single probabilistic model. This i.i.d. (independent and identically distributed) assumption significantly reduces analytical complexity. While it assumes independence, it remains applicable in practical scenarios where trajectories exhibit weak correlations, a common feature in many real-world systems. For instance, trajectories can be treated as independent samples from multiple agents or as repeated simulations of the same agent under varying initial conditions.

By adopting this assumption, we unify the analysis into a single macroscopic model: the stochastic process $\{\rho_k(x)\}_{k=0}^{\infty}$, which represents the evolution of probability densities over time. This abstraction allows us to study the system's collective behavior at a higher level, avoiding the need to track each particle or agent individually.

As shown in Section 1.3 and Lemma 1, for trajectories collection we consider in this paper, the density evolution chain $\{\rho_k(x)\}_{k=0}^{\infty}$ forms a Markov process (Markov chain). Next, we will introduce some basic properties of Markov processes (chains) and Markov operators.

**Definition 1 (Ash (2012); Lasota & Mackey (2013))** *In probability theory and statistics, a (discrete) Markov chain or Markov process is a stochastic process $\{\rho_k(x)\}_{k=0}^{\infty}$ describing the evolution of states $x \in \mathbb{R}^M$, in which the state at time instant $k$ depends only on the state attained in the previous event $x_{k-1}$. In operator theory, a Markov operator propagates densities as, that is $P \circ \rho_k(x) = \rho_{k+1}(x)$. Here, $P$ is a linear operator on a certain function space (positive $\mathcal{L}_1$ function space) that conserves the $\mathcal{L}_1$ norm (the so-called Markov property).*

In other words, for the corresponding Markov transfer operator $P$ that propagates the density $\rho_k(x)$ forward, we have that [Lasota & Mackey (2013)]

- $P$ is a linear operator;
- $P \circ \rho(x)$ is non-negative if $\rho(x)$ is non-negative;
- Integral invariance: For $\rho(x) \geq 0$,

$$\int P \circ \rho(x)dx = \int \rho(x)dx;$$

Many practical systems, especially those involving controlled stochastic dynamics, exhibit a special property known as constrictiveness. A constrictive Markov process is one that asymptotically converges to a group of periodic densities $\{g_i(x)\}_{i=1}^{l}$ [Lasota & Mackey (2013)]. It is obvious that this class of Markov chains have a stationary density $\rho^*(x) = \frac{1}{l} \sum_{i=1}^{l} g_i(x)$ and this stationary density satisfies the fixed-point equation [Lasota & Mackey (2013)]:

$$P \circ \rho^*(x) = \rho^*(x).$$

This stationary density represents the long-term behavior of the system, encapsulating its equilibrium state. For instance, in a controlled robotic system, $\rho^*(x)$ might describe the distribution of stable states achieved under a given control policy. The existence of $\rho^*(x)$ has profound implications:

- It provides a concrete representation of the system's asymptotic behavior.
- It enables assessment of system alignment with desired objectives. For example, a generative AI model trained to sample from a specific distribution can be evaluated by comparing its stationary density $\rho^*(x)$ to the target distribution.

This macroscopic approach to density evolution has broad applications in AI and robotics. By focusing on the evolution of probability densities, we can predict and analyze the behavior of complex systems without explicitly modeling individual components. For example:

- Robotics: Assessing the stability and reliability of control policies.
- Generative AI: Evaluating sampling processes in diffusion-based models.
- Crowd Dynamics: Understanding collective motion in human or animal groups.

Furthermore, this framework offers computational advantages. Directly tracking individual trajectories in high-dimensional systems is often infeasible due to the curse of dimensionality. Instead, modeling the evolution of densities provides a scalable and efficient alternative.

## B   HYPERPARAMETER OF PISA IN EXPERIMENTS

| Hyperparameters of PISA in Different Experiments | | | |
|---|---|---|---|
| Hyperparameter | Lunar Lander | Score-Based Generative Model | UCY Pedestrian |
| $\lambda$ | $5 \times 10^{-4}$ | $5 \times 10^{-4}$ | $5 \times 10^{-4}$ |
| $\mu$ | $5 \times 10^{-4}$ | $5 \times 10^{-4}$ | $5 \times 10^{-4}$ |
| $l$ | 5 | 5 (2, 10, 20 for comparison) | 5 |
| $N_{epochs}$ | 1000 | 1000 | 1000 |
| Optimization Method | Adam | Adam | Adam |
| Learning Rate | $5 \times 10^{-3}$ | $5 \times 10^{-3}$ | $5 \times 10^{-3}$ |
| $K$ | 100 | 500 | 600 |
| Hidden Layers of $A_\theta^i(\rho)$ and $G_\gamma^i(x)$ | 3 | 3 | 3 |
| Kernel used in KDE | Gaussian Kernel | Gaussian Kernel | Gaussian Kernel |
| Bandwidth | 1.0 | 0.8 | 0.6 |
| Number of Sample Trajectories ($N$) | 3000 | 5000 | 400 |
| Number of Reference Points ($N_{ref}$) for KDE | 3000 | 3000 | 3000 |

**Remark 4 (On the Role of Hyperparameters $\lambda$ and $\mu$)** *The hyperparameters $\lambda$ and $\mu$ in the cost function $L(\theta, \gamma)$ play a pivotal role in the PISA algorithm by governing the balance between short-term prediction accuracy and the correct estimation of terminal behavior. Specifically, these hyperparameters are associated with the following terms in the loss function:*

- *$\lambda \sum_{i \neq j} \langle G_i^\gamma(x), G_j^\gamma(x) \rangle$: This term enforces orthogonality among the learned basis functions $G_i^\gamma(x)$, which is crucial for the spectral decomposition of the Markov transfer operator. A higher $\lambda$ ensures better orthogonality, improving the ability to capture stationary behavior at the cost of possibly overfitting to long-term properties.*

- *$\mu \sum_{r=1}^{l} D\left(\sum_{i=1}^{l} A_i^\theta(G_r^\gamma) G_i^\gamma(x) \,\|\, G_{r+1}^\gamma(x)\right)$: This term captures the cyclic property of the Markov operator in its spectral decomposition. A larger $\mu$ emphasizes the role of capturing the long-term asymptotic behavior of the densities.*

*In this work, we choose relatively small values for $\lambda$ and $\mu$ ($5 \times 10^{-4}$) across all experiments, as detailed in the table. This choice reflects a higher emphasis on achieving accurate short-term predictions of the density evolution, as these predictions are critical for evaluating immediate system behavior in tasks like autonomous control, generative modeling, and human-robot interaction. However, such a choice represents a trade-off: while short-term predictions are improved, the accuracy in estimating the terminal behavior, including the stationary density, may be reduced.*

*Applications that require detailed insights into long-term stability or alignment properties, such as analyzing the terminal density of reinforcement learning agents or validating generative models, might benefit from larger values of $\lambda$ and $\mu$. Increasing these parameters enhances the algorithm's sensitivity to asymptotic properties of the Markov transfer operator but might reduce the fidelity of short-term density predictions.*

*The values of $\lambda$ and $\mu$ can thus be tailored to the specific requirements of the application at hand. For instance:*

- *In tasks where short-term predictions dominate the performance metrics (e.g., predicting pedestrian motion or robot trajectory planning), lower values of $\lambda$ and $\mu$ are preferable.*

- *For tasks focused on evaluating system alignment or ensuring stability over long time horizons, higher values may be more appropriate.*

972
973
974
975
976
977
978
979
980
981
982
983
984
985
986
987
988
989
990
991
992
993
994
995
996
997
998
999
1000
1001
1002
1003
1004
1005
1006
1007
1008
1009
1010
1011
1012
1013
1014
1015
1016
1017
1018
1019
1020
1021
1022
1023
1024
1025

*This flexibility allows the PISA framework to adapt to a wide range of applications by adjusting the relative weighting of short-term and long-term considerations in the loss function.*

