# OpenReview forum: "PREDICTING THE BEHAVIOR OF AI AGENTS USING TRANSFER OPERATORS"
_ICLR.cc/2025/Conference — Submitted to ICLR 2025_

### Official Review · Reviewer_h16B · 2024-10-24

**Soundness:** 2
**Presentation:** 2
**Contribution:** 2
**Rating:** 5
**Confidence:** 2

**Summary:**

The paper aims to predict the behavior of AI systems using density estimation and models of the stochastic dynamics of the systems. It introduces a method based on learning the evolution of probability densities from trajectory data. The probability densities at each time step are first non-parametrically approximated using kernel density estimates. A transfer operator on the densities is then learned assuming a particular proposed functional form involving neural networks. The method is evaluated on three examples: a reinforcement learning agent in a continuous control domain, a score-based generative model, and a dataset of pedestrian walking. In comparison with two baseline methods, the predicted densities are closer to the true densities in terms of the KL divergence.

**Strengths:**

In general, the paper tackles an interesting problem and the results are clearly presented. The key strength of the method seems to be the ability to directly predict the stationary distribution (eq. 9), which would not be straightforward to obtain for many other approaches.

**Weaknesses:**

Because the paper does not situate the proposed method in a broader context (neither by discussing other work on the problem of predicting AI systems' behavior, nor by discussing how the results compare to alternative approaches), I hard a hard time judging the contribution of this paper.

1. The introduction section contains basically no citations. Has nobody else worked on the problem of predicting the behavior of AI systems? There are several places where the text hints at relevant work, but does not cite any. Here are some examples (although this is not an exhaustive list):
    - "The integration of artificial intelligence (AI) models within autonomous agents has transformed many fields" (l. 26 - 27)
    - "there has been a notable increase in modeling these behaviors as nonlinear dynamical systems" (l. 46 - 47)
    - "techniques such as Dynamic Mode Decomposition (DMD) and its generalizations have demonstrated significant capability in revealing the underlying evolutionary laws of AI agents" (l. 47 - 48)
    - "Although the application of probabilistic models to learn and predict the statistical behavior of complex AI agents has increasingly attracted interest in areas such as autonomous driving, motion planning, and human-robot interaction" (l. 53 - 54)
2. The paper does very little to be accessible to a broader audience that might be in interested in predicting behavior but is not well-versed in statistical mechanics. In particular, a section providing a bit more mathematical background and intuition about the Perron-Frobenius operator (transfer operator / Markov operator) would be useful.
3. The part where the reasoning behind the model and loss function is explained (Section 4) would be more useful in the methods section instead of after the results.
4. The description of the methods is not very detailed, so that reproducing the results would be difficult just from the paper. For example, there is no indication of how the hyperparameters of the training were set, and little detail is provided on the neural network architectures and the training procedure. I understand that there are page limits, but there is also no code provided and no supplementary material.
5. No code is provided as supplementary material, which might have been helpful to address the shortcoming mentioned in the previous point and to get an intuitive understanding about how the abstract mathematical concepts are represented in concrete code.
6. As someone who was not familiar with this kind of modeling of the evolution of non-parametrically estimated densities, it was hard for me to judge how the approach compares to alternative appraoches, such as directly modeling the stochastic dynamics of the state (e.g. using a parametric model). This narrowness might be fine if the paper wants to provide a specific technical contribution in the context of these approaches. But the way the introduction is set up with the quite general goal of predicting the behavior of AI systems, I expected at least some comparison to alternative approaches. This applies to several sections of the paper
    a. The literature review is quite narrowly focused on methods for estimating the transfer operator. What about other approaches to reachability analysis, trajectory prediction etc.?
    b. The results only includes a comparison with two other methods. Are these the only applicable baseline methods? Please justify.
    c. The discussion section also does not set the method in a broader context. It hints at the limitations resulting from using KDE to approximate the densities. How does this compare to possible alternative approaches?
7. Formatting errors
    - Section 2.1 still contains parts of the instructions for using ICLR's Latex template (l. 180 - 181)
    - The citation style often makes no distinction between in-text citations and citations that should be in parentheses (e.g. l.60, l. 105 - 106, l. 153)

**Questions:**

- Is the method specific to the use case of predicting the behavior of AI systems or is it applicable in general to stochastic dynamical systems? The introduction suggests the former, but the rest of the paper the latter.
- What is the role of the two hyperparameters of the loss function ($\lambda$ and $\mu$)? How were they set? How does changing these hyperparameters affect the performance of the method?
- Algorithm 1: how are the constraints on the functions $G_\gamma^i$ and $A_\theta^i$ enforced? Which optimization algorithm is used to train the neural networks?
- Non-parametric density estimation techniques are known to be particularly prone to the curse of dimensionality. How does the method scale to higher-dimensional spaces?

---

> ### Comment · Reviewer_nU6Y · 2024-11-24
>
> I agree with reviewer h16B's points that some parts of the paper were hard to understand and were not easily accessible.

---

### Official Review · Reviewer_s1m7 · 2024-11-03

**Soundness:** 3
**Presentation:** 3
**Contribution:** 2
**Rating:** 5
**Confidence:** 4

**Summary:**

This paper introduces a method for producing the statistical behavior (terminal distribution) of agents using the  Fokker-Planck equation.

**Strengths:**

The paper is well-written and well-motivated.
The statistical analysis appears to be sound.
To the best of this reviewer's knowledge, this exact method for agent modeling has not been previously defined.

**Weaknesses:**

The paper seems to suggest that the idea of agent modeling originated in the 2020s.  All related work is from that time or later.
There is a survey entitled "Autonomous Agents Modelling Other Agents" that was published in 2017 and covers research from at least the two decades prior to that.  To properly assess the novelty of this approach, the authors should relate to the prior research and identify the closest methods for direct comparison.

Finding the "terminal distribution" of the agent behavior appears to amount to finding the stationary distribution of a Markov process.  Is that the case?  If so, there have been prior methods for doing so that ought to be compared.

If I understand correctly, the approach is designed for a purely single agent context, without any strategic interactions among agents.  Nonetheless, it is assessed in a pedestrian domain, which is an inherently multiagent setting.  Methods such as replicator dynamics (e.g. see the work of Karl Tulys et al.) could be brought to bear for finding the terminal distribution.

**Questions:**

Please see the weaknesses section.

---

### Official Review · Reviewer_nU6Y · 2024-11-08

**Soundness:** 3
**Presentation:** 3
**Contribution:** 3
**Rating:** 8
**Confidence:** 2

**Summary:**

The paper addresses the challenge of predicting AI agents' behavior, treating these agents as stochastic nonlinear dynamical systems. Using a probabilistic approach, the authors propose a framework based on the Fokker-Planck equation to predict statistical behaviors via an entropy minimization strategy. Their primary contribution is the PISA algorithm, which enables accurate predictions of agents' behavioral density evolution, particularly over long horizons. PISA leverages the spectral decomposition theorem to simultaneously approximate the Markov operator from agent trajectory data and predict asymptotic behavior. The authors demonstrate PISA's effectiveness in diverse applications, including robot trajectory prediction, generative model behavior, and pedestrian movement forecasting.

**Strengths:**

- The paper introduces a unique probabilistic perspective on AI agent behavior, combining concepts from stochastic processes with the Fokker-Planck equation. The originality stems from adapting a statistical density-based approach for complex, high-dimensional AI-driven environments. The spectral decomposition-based formulation for behavioral density evolution is novel in predicting long-term agent alignment.
- The mathematical rigor is evident, with clear derivations of the density evolution framework and detailed algorithmic steps. The PISA algorithm’s grounding in spectral decomposition provides robust theoretical backing. Although, it might be possible that I have not completely understood some parts of the proof.
- I feel that this research tries to address the need to understand and predict the behavior of complex AI agents, which has critical implications for fields requiring safety and reliability in autonomous systems. Applications like reinforcement learning, generative modeling, and pedestrian prediction show the method's versatility.

**Weaknesses:**

- While PISA demonstrates strong performance theoretically and in small-scale applications, its practicality in real-time, high-dimensional systems may be limited. The algorithm's scalability with respect to density estimation (e.g., kernel density estimation) needs clearer justification or further exploration in high-dimensional environments.
- The paper's assumption of Markov properties in agent dynamics may not always hold in certain AI-driven systems, such as those influenced by long-term dependencies or non-stationary environments. Additionally, relying on a fixed Gaussian kernel could introduce estimation bias, potentially underestimating density variations in non-Gaussian distributions, especially for high-variance AI behaviors.

**Questions:**

1. Could the authors elaborate on the feasibility of adapting PISA for real-time, high-dimensional AI systems? Additionally, has the choice of Gaussian kernel in KDE been optimized for different applications? Would adaptive kernel techniques enhance density estimation accuracy?
2. The authors mention the computational resources used for experiments but do not discuss performance time or computational trade-offs explicitly. Can the authors quantify PISA's computational efficiency compared to DPDD and Meng et al., especially in scenarios requiring frequent density updates?
3. How robust is PISA if the Markov assumption for AI agent dynamics is slightly violated? Could incorporating non-Markovian extensions or memory-enhanced operators enhance prediction accuracy for more complex behaviors?
4. The paper evaluates PISA primarily through KL divergence. Have other evaluation metrics been considered (e.g., likelihood estimation or out-of-sample testing)? It would be insightful to understand the robustness of PISA across various performance metrics.

---

### Meta-Review · Area_Chair_vtHe · 2024-12-29

**Metareview:**

The paper proposes a behavior prediction method that treats agents as stochastic nonlinear dynamical systems and uses the Fokker-Planck equation to predict the statistical behavior. The data-driven approach, named PISA, simultaneously approximates the Markov operator for predicting the evolution of agents and their terminal probability density. The method's effectiveness is demonstrated across various applications, including robot trajectory prediction, generative model behavior, and pedestrian movement forecasting.

The reviewers acknowledge the paper's novel probabilistic perspective. The use of spectral decomposition for behavioral density evolution is also seen as a strength. However, there are concerns regarding the clarity and accessibility of the paper, particularly for those not well-versed in statistical mechanics. Reviewers also noted that the initial literature review was incomplete, and that the paper did not engage sufficiently with related work. While these concerns were largely addressed in revisions, the paper in its current state is 20% over the maximum length (12 out 10 maximum pages).

**Additional Comments On Reviewer Discussion:**

The authors significantly improved the paper during the rebuttal, by providing a more comprehensive literature review, clarifying technical details, adding mathematical background, and including an appendix with hyperparameters. They also provided code for the experiments. Reviewers acknowledged that the authors had improved the paper by addressing some of the concerns regarding the completeness of the literature review and the clarity of the method. However, some reviewers felt that some points were not completely addressed, such as the discussion of alternative methods, the role of hyperparameters in the cost function, and the overall presentation.

At the end, the paper in the current state, still requires a significant update, given that it is over the allowed page limit. The scope of the change required to shorten it would call for another around of peer reviews.

---

### Decision · Program_Chairs · 2025-01-22

Reject